# Research on a Recognition Algorithm for Traffic Signs in Foggy Environments Based on Image Defogging and Transformer

**DOI:** 10.3390/s24134370

**Published:** 2024-07-05

**Authors:** Zhaohui Liu, Jun Yan, Jinzhao Zhang

**Affiliations:** 1State Key Laboratory of Automotive Simulation and Control (ASCL), Changchun 130025, China; 2College of Transportation, Shandong University of Science and Technology, Qingdao 266590, China; yanjun0709xz@163.com (J.Y.); zhangjinzhao202103@163.com (J.Z.)

**Keywords:** foggy environment, traffic sign recognition, Pix2PixHD, YOLOv5, transformer

## Abstract

The efficient and accurate identification of traffic signs is crucial to the safety and reliability of active driving assistance and driverless vehicles. However, the accurate detection of traffic signs under extreme cases remains challenging. Aiming at the problems of missing detection and false detection in traffic sign recognition in fog traffic scenes, this paper proposes a recognition algorithm for traffic signs based on pix2pixHD+YOLOv5-T. Firstly, the defogging model is generated by training the pix2pixHD network to meet the advanced visual task. Secondly, in order to better match the defogging algorithm with the target detection algorithm, the algorithm YOLOv5-Transformer is proposed by introducing a transformer module into the backbone of YOLOv5. Finally, the defogging algorithm pix2pixHD is combined with the improved YOLOv5 detection algorithm to complete the recognition of traffic signs in foggy environments. Comparative experiments proved that the traffic sign recognition algorithm proposed in this paper can effectively reduce the impact of a foggy environment on traffic sign recognition. Compared with the YOLOv5-T and YOLOv5 algorithms in moderate fog environments, the overall improvement of this algorithm is achieved. The precision of traffic sign recognition of the algorithm in the fog traffic scene reached 78.5%, the recall rate was 72.2%, and mAP@0.5 was 82.8%.

## 1. Introduction

In recent years, the occurrence and frequency of haze have constantly been increasing, bringing severe threats to people’s daily lives [1]. Foggy weather is one of the most common adverse weather conditions as it reduces visibility; it is always a major hidden risk to traffic safety. Traffic signs contain important semantic information such as road conditions, driving environment, and speed limit warnings, which play a significant role in reducing the rate of road accidents, therefore, it is vital to recognize these signs quickly and accurately [2].

Detection algorithms for traffic signs are now primarily divided into two categories: detection algorithms based on traditional methods and detection algorithms based on deep learning. Traditional detection algorithms for traffic signs include the detection method based on color features and detection methods based on shape features. Among them, the former detects traffic signs according to color models such as RGB, HSV, and HSI, while the latter is also the subject of extensive research conducted by scholars. J. M. LIillo Castellano et al. [3] studied the segmentation region methods based on HSV color models to extract traffic sign information after segmenting the specified regions. C. Bahilmann et al. [4] considered the color and Haar features of images, and achieved traffic sign detection using AdaBoost algorithms. C. G. KIRAN et al. [5] first performed the color segmentation of images and combined segmented chunks with edge features, and then used a support vector machine (SVM) to achieve traffic sign classification. However, since the selective methods of traditional features are subjective and easily affected by complex environments, deep learning-based traffic sign detection methods are more advantageous. This algorithm learns features by training a large amount of data, which is more accurate than the traditional methods that use artificially designed features, and its precision is significantly improved.

Currently, mainstream object detection algorithms are categorized into two-stage object detection algorithms and single-stage object detection algorithms. The Faster R-CNN [6] is a classic representative of the two-stage algorithm, which adds a region proposal network (RPN) on the basis of a Fast R-CNN, realizes CNN feature sharing, and uses a convolutional neural network (CNN) to automatically generate a region proposal instead of the traditional generating method for a region proposal, which improves the speed of network computation [7]. In deep learning-based object detection algorithms, the two-stage detection algorithm has high accuracy but a slow speed of recognition, so this kind of algorithm usually needs to be optimized in traffic sign detection scenes. In [8], the authors proposed a Fast R-CNN based on a residual network for multi-task parallel detection and applied it to road environment perception; they succeeded in detecting multiple types of targets with a detection speed of more than 7 fps. In [9], the authors mentioned that an improved classifier based on the Fast R-CNN and the recognition precision on the TT-100K datasets was increased by 7.7%, and the detection speed was also greatly improved. Jinghao Cao et al. [10] added an attention mechanism module to the backbone of a Fast R-CNN and incorporated multi-scale fusion architecture to improve the detection precision of traffic signs.

The end-to-end single-stage model mainly includes you only look once (YOLO) [11], the single-shot multi-box detector (SSD) [12], etc. Sun Fuwen [13] improved the YOLOv3 model by proposing an enhanced version and using reduced down-sampling feature maps (four times four) to predict target information. The experimental results showed that when small-sized traffic signs are recognized, the mean average precision (mAP) of an improved YOLOv3 model is 3.7% higher than that of the original YOLOv3. In [14], the authors combined synthetic images with original images through data augmentation to expand the dataset distribution and improve the detection performance. The precision in YOLOv3 and YOLOv4 is 84.9% and 89.33%, respectively. J. Wang [15] used the attention fusion feature pyramid network (AF-FPN) to replace the original FPN in YOLOv5. He improved the detection performance of the YOLOv5 network for multi-scale targets while ensuring real-time detection. Yin Jinghan et al. [16] optimized the architecture of the YOLOv5 network, adopted reverse thinking, reduced the depth-aware feature pyramid network (FPN), and limited the maximum down-sampling multiples to solve the problem of small targets being difficult to recognize. This optimization enables the network to maintain high-precision recognition even in adverse weather conditions.

There are generally two methods for traffic sign detection in a foggy environment. The first is to process foggy images and then perform detection. Among them, the most common method for processing foggy images is the defogging algorithm. He et al. [17] proposed an image-defogging algorithm based on the dark channel prior (DCP) theory to optimize the transmission estimation and effectively avoid color-shifting artifacts in the bright areas of an image and the halo effect in the sudden transition of the depth of field. Cai et al. [18] at the South China University of Technology also designed the DehazeNet defogging algorithm based on CNN architecture. The occurrence of Pix2Pix [19] and Cycle GAN [20] enabled the conversion of one scene representation into another. Inspired by the above-mentioned generative adversarial network (GAN), the authors in [21,22,23] used priors, which are different from the defogging algorithm; they learned the difference between fog and sunny day images through the adversarial network and realized the conversion of foggy images to sunny day images. The above-mentioned defogging algorithms are mostly used in low-level vision tasks, and there is still much room for improvement when facing high-level vision tasks (such as traffic sign recognition). The second method for detecting traffic signs in a foggy environment is to improve the robustness of the target detection network in the foggy environment by optimizing the detection network structure. Pengfei Hu [24] combined an improved DCP algorithm with an optimized LeNet5 network to achieve traffic sign recognition in a foggy environment, with a significant performance improvement but it was not validated using fog datasets. Shubo Yu [25] compared the performance of HOG-SVM and improved the Faster RCNN in fog and found that HOG-SVM is limited in complex scenes, while the Faster RCNN improves accuracy and detection, but the validation set contained fewer real fog images. Binke Lang et al. [26] improved YOLOv5 detection for traffic signs in foggy environments by introducing the BIFPN and CA, but mAP@0.5 only increased by 0.2% and lacked verification in real fog scenes. In [16,24,25,26,27,28,29], the authors used an FPN as the backbone of a multi-level feature pyramid to better detect small targets and achieve good detection results even in complex environmental backgrounds. In addition to using a parallel pyramid network to improve feature extraction, [16] introduced a residual aggregation network. In [30], the authors extracted detailed features by building a residual aggregation module, using dense connections, and combining low-dimensional features to generate high-dimensional features. However, these methods could not fundamentally solve the impact of a foggy environment on the visibility of traffic signs.

Based on the aforementioned research, in traditional defogging algorithms, both image enhancement methods and image restoration methods lead to the loss of feature information of small targets in images; in deep learning-based defogging algorithms, the use of a neural network to estimate parameters such as atmospheric transmission also lead to estimation errors. GANs are used to remove fog and can circumvent the uncertainties of prior information, such as estimating atmospheric light intensity, but in the process of generating object-style images, there are still problems with the loss of local feature information and domain transfer. Improving the target detection algorithm usually involves changing the number and size of the convolutional network and adding an attention mechanism to improve the target detection algorithm’s ability to extract target features. However, increasing the receptive field by stacking convolution kernels makes it fundamentally difficult to solve the problem of information loss during feature extraction. Inspired by [31], this paper leverages the dynamic game characteristics of the discriminator and generator in the conditional generative adversarial network and applies the Pix2PixHD network to the task of defogging traffic sign images in foggy traffic scenes. In order to better match the defogging algorithm with the object detection algorithm, as well as improve the speed and accuracy of detection, the feature extraction module in YOLOv5 is improved. The transformer encoder module is integrated into the YOLOv5 detection model, which introduces a multi-head attention mechanism to establish connections between the parts of target images, thus reducing the influence of a foggy environment on traffic sign detection. At the same time, the iterations of the convolutional module are reduced, and the training efficiency of the model is enhanced so as to improve the detection speed of traffic signs. Finally, the defogging algorithm based on the Pix2PixHD network is combined with the improved YOLOv5 model for the recognition of traffic signs in foggy environments, and the effectiveness of the proposed method is verified using comparative experiments.

The contribution made in this paper mainly involves aiming to solve the difficult problem of identifying traffic signs as small targets in foggy traffic scenes. First, the conditional generation adversarial network Pix2PixHD is trained, the changing trend of its network loss function is visualized, and a qualitative and quantitative comparison with mainstream defogging algorithms is carried out, which proves the applicability and effectiveness of Pix2PixHD for defogging traffic sign images in foggy traffic scenes. Then, the defogging algorithm based on the Pix2PixHD network is used to defog traffic sign images in a foggy environment and then obtain the conversion of the foggy image to a clear image, which provides a basis for the selection of the defogging algorithm of the traffic sign images in the foggy traffic scene. It effectively avoids generating estimation errors in the atmospheric light intensity and transmission rate and the problem of dimension conversion. At the same time, it generates high-definition traffic sign images conducive to the recognition of the detection algorithm. Secondly, a transformer is used to improve both the shallow network and deep network in the feature extraction network of YOLOv5 Backbone, and the YOLOv5-T detection algorithms is proposed, which effectively improves the extraction ability of the network to the features of the traffic sign image. Finally, the image-defogging algorithm based on Pix2PixHD and the YOLOv5-T detection algorithm are combined to realize the identification of traffic signs in a foggy environment. The comparative experiments demonstrated that the method proposed in this paper is significantly better than other methods, especially in moderate fog datasets. The overall research idea of this paper is shown in Figure 1.

## 2. Principle of the Defogging Algorithm Based on the Conditional Generative Adversarial Network

A conditional generative adversarial network (CGAN) is often used for scenarios such as semantic segmentation [31] and image enhancement. In this paper, the dynamic game characteristics of the discriminator and generator in a CGAN are utilized for the task of defogging traffic sign images in foggy traffic scenes.

### 2.1. Principle of the Conditional Generative Adversarial Network (CGAN)

The CGAN is roughly similar to traditional GANs in structure, consisting of a generator (G) and a discriminator (D). A CGAN adds a constraint condition y to the GAN, as shown in Figure 2. The input foggy image x is fed into the generator, which continuously learns the feature distribution of sunny day images y to produce the generated image G(x). Based on the input actual paired images y and x, the discriminator judges G(x) and the input foggy image x whether they satisfy the requirements of the objective function. The training process of the CGAN involves a dynamic “zero-sum game” between G and D. The discriminator evaluates the adequacy of the generated image based on the input labeled image and then provides feedback to the generator through backpropagation. Through continuous iterations between G and D, G(x) gradually approaches the real image y. The ultimate goal of the model is to make D unable to distinguish between the authenticity of the generated data G(x).

### 2.2. Pix2PixHD Network

#### 2.2.1. Architecture of the Pix2PixHD Generator

The Pix2PixHD model is an improvement based on the CGAN. Its core principle is the same as the principle of the CGAN, and its architecture can be seen as an integration of two CGANs with different sizes. As shown in Figure 3, the multi-scale generator of Pix2PixHD consists of two modules, G1 and G2, which are similar in structure. G1 represents the global generator network, with an input and output image size of 1024 × 512. G2 represents the local enhancer network, with an input and output image size of 2048 × 1024. Overall, another generator (G1) is embedded in a regular generator (G2). The global structural features extracted by G1 and the features extracted by the shallow layer of G2 are fused and input into the middle layer of G2 to generate high-quality images. During the training process, G1, with a smaller resolution, is first trained, and then G1 and G2 are trained together. The input fog image is double down-sampled via the convolution layer of generator G2, and then another generator, G1, is used to generate low-resolution images. The output generated by G1 and the image obtained from the down-sampling carry out element-wise adding, and then this combined result is fed into the subsequent network of G2 to generate high-resolution images. In the generator structure of Pix2PixHD, double down-sampling refers to reducing the width and height of the input image to half of its original size through a convolutional layer with a step size of two in order to extract the features of images on different scales and reduce the computational load for subsequent processing.

#### 2.2.2. Architecture of the Pix2PixHD Discriminator

The Pix2PixHD discriminator is an improvement based on the Markovian discriminator (PatchGAN). The architecture of PatchGAN is illustrated in Figure 4. First, the discriminator utilizes a convolutional neural network (CNN) to divide the input images into multiple patches and perform continuous down-sampling. A patch of the feature map extracted using the high-level CNN corresponds to an area of the low-level feature map, representing a receptive field. Then, the discriminator is used to make a decision for each patch of the output layer image. Finally, the average loss of all patches on the entire image is taken as the final result.

To generate higher-resolution images, it is not enough for Pix2PixHD to improve the generator alone. The discriminator also needs to be optimized. First, the discriminator needs to have a large receptive field, so a deeper network is also needed. However, stacked convolutional layers can lead to overfitting and require significant memory usage. Based on PatchGAN, the multi-scale discriminator has been proposed, which employs three discriminators D1, D2, and D3 with the same network architecture but operating on different image scales. An image pyramid comprising three scales is constructed by down-sampling the real and synthetic high-resolution images by factors of 2 and 4. Specifically, D1 operating on the coarsest image (the one with the lowest resolution) has the largest receptive field, enabling it to understand the image more globally and provide feedback to the generator to produce globally consistent images. In contrast, D3 operating on the finest image (the one with the highest resolution) is responsible for guiding the generator to generate details. D2 lies between these two, offering an intermediate scale evaluation that focuses on both details and the overall integrity of the image. By adopting a multi-scale discriminator, Pix2PixHD is able to evaluate different aspects of the generated image better, thereby enhancing the fidelity and quality of the image generation.

## 3. Training Experiments Based on the Pix2PixHD Network

### 3.1. Building the Training Dataset

To utilize GANs for defogging traffic sign images in foggy environments, a large number of paired image samples are required for training. However, in reality, it is difficult to collect a large number of traffic sign images in a foggy environment as well as their corresponding sunny day images. Therefore, open-source datasets and captured foggy images were used to construct the dataset of this paper. The TT100K dataset [32] was used for the sunny day traffic sign dataset. This dataset was derived from a Chinese street view containing more than 300 cities with rich scenes and complete types of traffic signs. To expand the sample of the dataset, the sunny day images were fogged according to the real fog images.

To make the synthetic foggy traffic scenes as close as possible to real foggy scenes, MATLAB was used to simulate light fog and medium fog conditions based on the principle of the atmospheric scattering model [33]. Specifically, the atmospheric light value A was set to 0.8, and the scattering coefficient β was set to 0.04 and 0.08. The construction process of the synthetic foggy images is shown in Figure 5. Some images of the dataset constructed in this paper are shown in Figure 6.

### 3.2. Training Parameter Configuration

The debugged Pix2PixHD model was trained by using Pytorch 1.10.0+cuda 11.3. The training batch size was set to 1, the initial learning rate was set to 0.0002, and the number of epochs was 300. The detailed parameters are presented in Table 1.

### 3.3. Analysis of the Training Results

The loss function curve is a crucial indicator in the training of deep-learning models. It can reflect the changes and training results of the model during the training process. By observing the loss function, we can clearly understand the stability of the model training process, whether the model has achieved optimization, and the quality of the final model’s performance. The visualization tool TensorBoard was used to visualize the changing trend of the loss functions of the discriminator and generator during Pix2PixHD training. To fully prove the convergence of each loss function, we visualized all 300 epochs. As can be seen from Figure 7, although the loss function of both the discriminator and the generator had some oscillations, they show an overall downward trend. At the same time, the convergence rate was the fastest in the first 100 epochs, and after the 250th epoch, the value of the loss function hardly changed. This shows that the model continuously learned the characteristics of sunny days and foggy environments, and the performance of the model gradually stabilized after a period of training. Among them, D is the changing trend of the loss function of the discriminator, and G is the changing trend of the loss function of the generator.

### 3.4. Comparative Analysis of the Defogging Effect

To ensure the accurate and efficient recognition of traffic sign images in foggy traffic scenes, defogged images should not only guarantee the clarity of the global image but also require sufficient clarity of local targets. To verify whether the defogging algorithm based on the Pix2PixHD network used in this paper could meet this requirement, we introduced the CAP [34], DCP [35], Dehaze-Net [18], FFA-Net [36], GCA-Net [22], Cycle GAN [23], and AOD-Net [37] defogging algorithms for a comparative analysis from both qualitative and quantitative aspects.

#### 3.4.1. Qualitative Analysis

The defogging algorithm proposed in this paper was used to process traffic sign images in foggy traffic scenes, and a visualization comparison was made with other defogging algorithms. As shown in Figure 8, the blue wireframe shows the generated foggy images and the corresponding clear sky images; the green wireframe shows the defogging effect of the compared defogging algorithms and the defogging algorithm proposed in this paper on the foggy images; and the red wireframe shows the traffic sign that needs to be recognized. Through observation, it can be found that the traffic sign features in the image processed using the defogging algorithm presented in this paper are significantly enhanced, and the overall clarity of the image is also greatly improved. The traffic sign features are noticeable after defogging with the DCP method, but there is a noticeable distortion in the overall image. While the Cycle GAN achieves good global defogging, it suffers from severe loss of traffic sign feature information. After defogging with the FFA-Net, the clarity of large objects in the image is indeed improved, but small objects remain blurry. CAP adds other noise while defogging, making the image more blurred, indicating that this algorithm is not suitable for traffic scenes. The enhancement effect of the image feature of the other algorithms is not obvious. The above-mentioned comparison shows that the Pix2PixHD defogging algorithm is not only suitable for traffic scenes but also has great advantages in processing image details and is suitable for restoring traffic signs in a foggy environment.

#### 3.4.2. Quantitative Analysis

At present, there is no unified evaluation standard for the quality of image defogging. In this paper, two common image quality evaluation indexes, the peak signal-to-noise ratio (PSNR) and structural similarity (SSIM), were selected to evaluate the defogging quality. The PSNR is a commonly used indicator for the objective quantitative evaluation of image quality, which is used to evaluate the errors between the corresponding pixels of two images. The larger the PSNR value is, the better the image-defogging effect is. The SSIM is an indicator for measuring the similarity between two images, ranging from 0 to 1. The larger the SSIM value is, the higher the similarity is. When the two images are exactly the same, the SSIM is one. Compared with the PSNR, the SSIM is more consistent with the judgment of image quality from the perspective of human eyes. The traffic sign images in light fog and medium fog were tested separately. The highest index is marked in red font, and the lowest index is marked in blue font. The testing results are shown in Table 2 and Table 3.

As shown in Table 2, in a light fog environment, the SSIM value of Dehaze-Net is 0.898, and the SSIM value of Pix2PixHD is 0.897, with a difference of only 0.001. However, the PSNR value of Pix2PixHD is 24.269, and the PSNR value of Dehaze-Net is 22.520. In a medium fog environment, the SSIM and PSNR values of Pix2PixHD are both the highest, being 0.975 and 26.465, respectively. The structural similarity is 0.04~0.7 higher than the other defogging algorithms.

Combined with the results of the qualitative analysis, although the SSIM and PSNR values of FFA-Net, Dehaze-Net, and DCP are also high, the visualization effects of these three defogging algorithms are average. This indicates that while these three defogging algorithms indeed reduced the influence of fog on images to a certain extent, they are not sufficient in reducing noise in the targets within the images in terms of overall style. Therefore, although the SSIM and PSNR values are not bad, the visualization effect is poor. During qualitative analysis, DCP and Cycle GAN exhibit different characteristics. The DCP algorithm performs well in restoring image details, but the overall color background is altered. On the other hand, Cycle GAN achieves good overall defogging effects but suffers from a significant loss of specific targets in the image, resulting in lower SSIM and PSNR values. Other defogging algorithms perform relatively poorly in both qualitative and quantitative analyses, indicating that these algorithms are not suitable for application in traffic scenes.

Based on the qualitative and quantitative comparative analysis results presented above, it is evident that the defogging algorithm based on the Pix2PixHD network demonstrates the best effect. The algorithm’s applicability and effectiveness for traffic signs in foggy traffic scenes are validated.

## 4. The Detection Algorithm for Traffic Signs Based on the Improved YOLOv5 Algorithm

### 4.1. Building the Dataset

The construction of the foggy traffic sign dataset was completed through image acquisition, image screening, and the labeling of traffic sign detection targets in the images. The sunny day dataset was selected from the TT100K dataset and the GTSDB dataset [38]. The foggy images were captured using a vehicle-mounted camera in urban driving scenes in Qingdao, including images of varying foggy conditions. In addition, to enhance the diversity of the data, foggy traffic sign images from other publicly available autonomous driving datasets on the internet, such as BDD100K [39], Oxford RobotCar Dataset [40], ApolloScape [41], were also utilized. These alternative datasets contained a total of 1000 foggy images. Furthermore, to meet the data requirements for model training, it was necessary to conduct statistics and screening on the number distribution of traffic signs in the dataset.

As shown in Figure 9, after data augmenting and screening, 45 types of traffic signs were collected. Among them, il100, p19, w32, etc., appeared less than 100 times and were not frequently used. Since a CNN cannot fully learn the characteristics of these traffic signs, it is necessary to further augment and screen the collected images. The screening results are shown in Figure 10.

Python was used to enhance the dataset of traffic sign images in this paper. Horizontal or vertical flipping, rotation, scaling, cropping, shearing, translation, contrast, color jitter, noise, and other operations were also used to simulate the overexposure, underexposure, and fog that traffic signs may experience in adverse weather conditions, thereby enhancing the robustness of the model in adverse weather conditions. The effect of traffic sign image augmentation is shown in Figure 11.

Finally, based on the important semantic information and the distribution of traffic sign quantities, a total of 30 categories of traffic signs were selected from the constructed dataset, including directive signs, warning signs, and prohibitory signs, along with their corresponding annotations, as shown in Figure 12. In traffic signs, blue is predominantly used for mandatory signs, such as “Straight Ahead”; red is typically employed for prohibitory signs, like “No Entry”; while yellow is mainly utilized for warning signs, such as “Pedestrians Crossing”.

### 4.2. Traffic Sign Detection Based on YOLOv5

As shown in Figure 13, the architecture of the YOLOv5 model is divided into three parts: the backbone, neck, and head. The backbone is the main structure of YOLOv5, which contains a CNN composed of convolutional modules such as Focus, Conv, and C3, mainly used to perform image enhancement and normalization on the input images. Spatial pyramid pooling (SPP) [42] was used to integrate local and global features, thereby improving the ability to represent feature maps. The backbone was used to extract feature information of the target images and input them into the next network layers. The neck part contains a series of convolutional layers that mix and combine image features, the core of which is the feature pyramid network (FPN) and path aggregation network (PANet). These structures can complement the features of various types of targets and enhance information transmission. They can accurately retain spatial information, help to locate pixels to form masks properly, and finally pass the fused image features to the prediction layer. The head is the detection architecture of YOLOv5. Traditional neural networks input the highest-level features into the detection layer, which causes the serious information loss of small target features after layers of convolution, making the target difficult to recognize [43]. YOLOv5 inputs the features extracted with the convolutional layers of different depths into the detect module and sets three detection heads for large, medium, and small target images to overcome the disadvantage of small target information loss.

On the basis of the dataset of 7000 images, the dataset was divided into a training set, a validation set, and a test set according to the ratio of 8:1:1. The training epoch value was 200. To highlight the influence of fog on traffic sign recognition, the trained YOLOv5 model was used to detect traffic signs in sunny and medium fog environments, and then the detection results were analyzed. The results are shown in Table 4. The foggy environments influence the recognition precision, recall, mAP@0.5, and mAP@0.5:0.95 of the model. In the medium foggy environment, the recall decreased by 15%, and mAP@0.5 and mAP@0.5:0.95 decreased by 14.2% and 13.6%, respectively.

### 4.3. Improvement of the YOLOV5 Target Detection Algorithm

There are some challenges in traffic sign detection. First, there is the problem of data imbalance, which includes the distribution of traffic sign types in the dataset being unbalanced and the number of positive and negative samples being unbalanced when the algorithm generates candidate boxes. Second, when a convolutional network extracts features from traffic signs, excessive convolutional layers can lead to the loss of feature information. To solve the imbalance problem of traffic sign data, the focus module performs data augmentation on the input images to ensure that the entire model has sufficient data for training. However, the extraction of detailed semantic information in traffic signs requires stacking enough convolutional layers, which causes the feature information of the detection target to be lost as the convolutional modules are stacked. Although the introduction of residual connections can improve the extraction of feature information, the computational complexity of the entire model also increases accordingly. A transformer [44] does not need to stack convolution modules to obtain global information like CNNs continuously. Instead, it can directly obtain global information by dividing the image into several regions [45]. However, a traditional transformer needs to construct many patches, and embedding sequences that are too long consumes a lot of computing power when calculating attention. The Swin transformer [46] solves this problem by using the principles of windows and layers.

#### 4.3.1. Principles of Transformer Architecture

The transformer is a new neural network architecture based on the self-attention mechanism. As shown in Figure 14, the input image is converted into an embedding vector. X∈ℝn×d, query (Q∈ℝn×dk), key (K∈ℝn×dk), and value (V∈ℝn×dv) represent the weight vector obtained after encoding. Query is used to inquire about information between other vectors, key is used to receive information about other input vectors, and value is the characteristic information of each input vector itself. The inner products of query and key are used to represent the correlation between the input vectors. To prevent the inner product from increasing with the increase in the dimension, the inner product of the two vectors (query and key) is divided by dk, normalized using the softmax function, and then the inner product is summed with value to obtain the attention value. In the attention value, the more relevant the target is to the detection target, the greater the proportion of its feature information. The attention value of each target in the target image is calculated in this way, and then weighted integration is performed to obtain the final weight of the self-attention mechanism. The attention value of each target in the target images is calculated in this way, and then weighted integration is performed to obtain the final weight of the self-attention mechanism, where n is the sequence length and d, dk, dv represent the dimensions of the vectors. For details, see Equation (1). The attention mechanism is the main module that constitutes the transformer. As shown in Figure 14, the transformer encoder inputs the image and extracts the image features. First, the image is divided into several regions (patches), and then each patch is encoded and embedded. After the embedded patches are normalized, they are inputted into the multi-head attention mechanism. The structure and workflow of the multi-head attention mechanism are the same as the above-mentioned attention mechanism, but this is a module in which the attention weights of different parameters are connected in parallel. It is then normalized and then enters the multi-layer perceptron (MLP). The entire structure also has two residual links to ensure the stable performance of the transformer module during multiple cycles.
(1)Attention(Q,K,V)=softmax(QKTdk)V

#### 4.3.2. Feature Extraction of Traffic Signs Using the YOLOv5 Backbone

The main task of the YOLOv5 backbone is to extract features from the input images and input the extracted features to the next module for feature fusion and detection. Therefore, feature extraction plays a decisive role in the final detection result. As shown in Figure 15, the input images first pass through the focus module to complete data enhancement and then pass through a series of convolution modules to extract the feature map. Among them, the feature maps extracted by the first C3_9, the second C3_9, and the last C3_3 need to be input into the next module in YOLOv5. However, with the increase in convolution operations, the extracted features become more complex, and some feature information is lost. In adverse weather conditions such as fog, snow, and rain, the captured traffic images contain a lot of noise [47], which also interferes with feature extraction. As mentioned above, the main reason why the YOLOv5 model has a poor effect on traffic sign detection in a foggy environment is that the backbone of YOLOv5 cannot effectively extract target feature information.

#### 4.3.3. Feature Extraction of Traffic Signs Using the Transformer

The transformer consists of a self-attention mechanism. Using a transformer encoder can enhance attention to small targets, improve the network receptive field, and reduce the computational and storage costs of low-resolution feature maps. The class activation mapping (CAM) method was introduced to visualize the process of extracting traffic sign features using a transformer and CNN and analyze the degree of attention paid to the same detection target with the model before and after the improvement. We can see that when using a transformer to extract features from traffic sign images, the attention mechanism can automatically match the detection target and reduce the degree of missed detection in target detection, which is something that conventional CNNs cannot do, as shown in Figure 16.

#### 4.3.4. Improvement of the YOLOv5 Network

In the original YOLOv5, the input images are continuously convolved through the various convolution modules of the backbone to extract features, and the extracted features are passed to the neck of YOLOv5. The neck module mainly uses the PANet structure to generate a feature pyramid to enhance the model’s detection ability for targets of different scales so that it can recognize the same target of different sizes and scales. The feature pyramid fuses the high-level features extracted using the convolution module with the low-level features, thereby improving the effect of target detection.

Traffic sign detection in foggy environments is not only affected by small size, but also by the foggy environment. When passing through layers of convolutional modules, feature information is lost, especially the feature information extracted using the last convolutional module of the backbone. The feature information output from the last layer is used for small target detection, so traffic sign detection in a foggy environment becomes very difficult.

To improve target feature extraction, target detection algorithms either use a combination of attention mechanisms and convolutional layers or change the number of neural network layers and the size of convolutions [46]. The transformer module is a module composed of a self-attention mechanism, which has great advantages in image feature extraction. First, the transformer module has a more complex feature representation capability and can better extract image features. Second, the model structure of the transformer module has better scalability, which can more easily expand the depth and complexity of the network and capture more feature information in the input images. In addition, the transformer module has better long-distance dependency modeling capabilities, can better model complex targets or scenes, and improve network performance. In view of the limitations of the feature information extraction of traffic signs in a foggy environment, this paper used transformer encoder blocks to replace some convolution blocks and C3 in the original YOLOv5 and used a transformer encoder as an independent feature extraction module to complete feature extraction and pass the extracted features to the head for detection. Each transformer encoder block contains two sublayers: the first sublayer is a multi-head attention layer, and the second sublayer (MLP) is a fully connected layer, and residual connections are used between each sublayer. The transformer encoder block increases the ability to capture different local information, and can also use the self-attention mechanism to explore the potential of feature representation. To ensure the improvement of the performance of the entire network models, the authors referred to the residual network (PANet) [48] and used the PANet to fuse the features extracted using the convolution modules of each layer with the features extracted using the transformer encoder module. The improved model is YOLOv5-T, and the structure is shown in Figure 17. Compared with the original bottleneck blocks in CSPDarknet 53, the YOLOv5 model with the transformer encoder module added at the end of the backbone can capture global information and rich contextual information, and reduce the computational cost and storage cost of high-resolution feature maps.

#### 4.3.5. Analysis of the Training Results

As shown in Figure 18, the loss values of both models are relatively large in the initial stage of training. As the number of training epochs increases, the loss values gradually decrease and eventually stabilize, which proves that both models converged after training. The loss value of the model YOLOv5-T after the introduction of the transformer encoder module converges faster than the original YOLOv5 model, and there are no large spikes. When the training reaches 200 epochs, the loss value reaches the lowest point, and the trained model performs well. When the improved model is used for training data, the loss function can achieve the same decrease effect in 100 epochs as that of the original model in 150 epochs, which fully proves that the use of the transformer encoder module requires fewer computing resources and has high computing efficiency to extract the feature information of the traffic sign.

In the precision–recall curve, the vertical axis is precision, and the horizontal axis is recall. The blue curve represents the total P–R curve for all categories, and the black curve represents the P–R curve for each category. The black curve close to the upper right corner indicates that the precision and recall of the traffic sign of this category are close to one in training, and the training effect is good; the black curve close to the lower left corner indicates that the precision and recall of the traffic sign of this category are low, which is mainly because the number of training samples for this type of traffic sign is low. The area enclosed by each black curve represents the mAP@0.5 value of each type of traffic sign, and the area enclosed by the blue curve represents the total mAP@0.5 value of the 30 types of traffic signs. As shown in Figure 19, in the data trained with the YOLOv5-T model, most of the black curves are located above the blue curve near the upper right corner, while when the original YOLOv5 model is used to train the same dataset, we can see that the black curves are scattered on both sides of the blue curve, with more near the lower left corner. The total mAP@0.5 of the improved model training is 0.805, while the original model mAP@0.5 is 0.758. Thus, mAP@0.5 significantly improved.

## 5. Recognition of Traffic Signs in Foggy Environments and Contrastive Analysis of the Recognition Effect

### 5.1. Influence of Fog on the Image Recognition of Traffic Signs

In foggy weather conditions, light is absorbed and refracted, resulting in the visibility of impaired visibility images. To analyze the specific influence of a foggy environment on the detection targets, a three-channel color histogram was used in this paper to count the pixel value distribution of traffic signs on sunny days and foggy environments. To reduce the influence of the background environment on the testing targets and consider the distance factors at the same time, the detection image was masked with the help of a Python script. As shown in Figure 20, in the same scene, the RGB three-channel histogram of the foggy images changed greatly. First, the actual pixel value of the image changed significantly. The peak values of the three colors of the foggy image increased significantly, which is twice the pixel value of the sunny image. Second, the peak distribution area of the red, green, and blue curves became smaller and concentrated together, and the brightness of the three peaks has an overall rightward shift trend. At the same time, the fluctuations of the red, green, and blue colors in the low brightness range changed from curves to straight lines. From the RGB histogram of the foggy images, it can be concluded that the overall brightness of the red, green, and blue colors in the image increased under the influence of foggy weather conditions, but the brightness distribution was concentrated. Colors with lower brightness disappear directly under the influence of fog. By combining these with the foggy scene images, we can see that the characteristics of foggy scene imaging are also expressed through the RGB three-channel histogram. Through the above analysis of the RGB histogram, we can see that there is the most obvious difference in color saturation between the sunny images and foggy images. In a sunny environment, due to sufficient light and bright colors, the color saturation of the image is relatively high, so the histogram distribution of its RGB three channels is relatively uniform. In a foggy environment, due to a large amount of water vapor and suspended matter that affect the penetration and propagation of light, the color saturation of the image decreases, the histogram distribution is relatively concentrated, and it presents a trend of being “thin and tall”. Compared with the sunny environment, the pixel values of the three channels of the detection targets in the foggy environment are concentrated at 200. This shows that the overall brightness of the images in the foggy environment improved, the pixel values in other intervals were reduced, and the distribution of the pixel values of each channel was unbalanced. Since traffic signs are composed of specific colors and shapes, the concentrated distribution of pixel values in each color channel in a foggy environment affects the detection of traffic signs.

### 5.2. Comparative Analysis of the Visualization of Traffic Sign Recognition Based on a Defogging Algorithm

In actual road scenes, traffic signs are small targets compared to other detection targets, so missed detection, false detection, and low precision are common problems. In response to the aforementioned problems, the main task of this paper is to improve the precision and efficiency of traffic sign recognition in a foggy environment by combining the defogging algorithms based on the Pix2PixHD network with the improved YOLOv5 algorithm, reflecting the comprehensive advantages of the combination of the two in the recognition of small targets such as traffic signs in actual road scenes. To fully verify the advantages of the Pix2PixHD+YOLOv5-T algorithms proposed in this paper, a comparative analysis of the visualizations of the recognition effects of synthetic foggy images of traffic signs and real foggy images of traffic signs in actual road scenes was carried out.

Figure 21 shows the experimental results of the comparative analysis of synthetic foggy images of traffic sign recognition in actual road scenes. We can see from this figure that the most obvious difference is that there is a big difference in the degree of missed detection of traffic signs. By using GCA-Net+YOLOv5-T and AOD-Net+YOLOv5-T for traffic sign recognition, GCA-Net and AOD-Net remove a lot of the target information in traffic scene images, which ultimately makes it unable to recognize traffic signs. When CAP+YOLOv5-T is used for traffic sign recognition, CAP does not reduce the noise caused by foggy weather conditions but increases noise, which makes it difficult for YOLOv5-T to recognize traffic signs. When CycleGan+YOLOv5-T is used to detect traffic signs, although CycleGan has an obvious defogging effect, it causes a serious loss of target details. Therefore, the CycleGan+YOLOv5-T algorithm is not suitable for the recognition of small targets such as traffic signs. The three algorithms FFA-Net+YOLOv5-T, Dehaze-Net+YOLOv5-T, and DCP+YOLOv5-T succeed in detecting two traffic signs, which outperform YOLOv5-T used directly in the process of recognizing a foggy environment. The algorithm Pix2PixHD+YOLOv5-T in this paper can recognize all traffic signs in the images, and its detection efficiency is significantly better than the other algorithms; at the same time, its precision is also higher than the other algorithms.

Figure 22 shows the experimental results of a comparative analysis of traffic sign image recognition in real foggy weather conditions. We can see from this figure that the Dehaze-Net+YOLOv5-T and DCP+YOLOv5-T algorithms have difficulty recognizing traffic signs in images of real foggy scenes due to the severe distortion of image brightness, color, and contrast after defogging. Although the GCA-Net+YOLOv5-T and FFA-Net+YOLOv5-T algorithms recognized the short-distance traffic sign No p11 (no honking), they failed to recognize the smaller traffic sign i5 (keep right) in the distance. This algorithm can be used to recognize large and short-distance targets, while traffic signs on the road are not only affected by their own size but also by distance. Therefore, the GCA-Net+YOLOv5-T and FFA-Net+YOLOv5-T algorithms are difficult to apply to the recognition of traffic signs in real road scenes of foggy environments. The Cycle Gan+YOLOv5-T algorithm did not detect the small traffic sign i5 (keep right) in the distance. At the same time, the close traffic sign p11 (no honking) was misidentified as p26 (no trucks allowed), resulting in serious misdetection. The AOD-Net+YOLOv5-T and CAP+YOLOv5-T algorithms did not recognize the traffic signs in the image; therefore, they are difficult to apply in the traffic sign recognition of road scenes in real foggy environments. The Pix2PixHD+YOLOv5-T proposed in this paper could not only recognize short-distance traffic signs but could also recognize long-distance small-target traffic signs in a real foggy environment. From the above comparative experimental results, it is safe to say that this paper improves the precision and recall of traffic sign recognition in a foggy environment by combining the Pix2PixHD network defogging algorithm with the improved YOLOv5 detection algorithm. Whether in synthetic foggy scenes or real foggy scenes, it can more effectively complete the task of traffic sign recognition.

### 5.3. Comparative Analysis of Three Recognition Methods for Traffic Signs in Foggy Environments

To further prove the advantages of the algorithm proposed in this paper, traffic sign images on sunny days and on real fog days were used as testing sets to conduct further comparative analysis of the various algorithms in this paper. The comparative experimental results show that the Pix2PixHD+YOLOv5-T algorithm is better than YOLOv5 and YOLOv5-T for recognizing traffic signs in a real foggy environment. The specific results are listed in Table 5.

We can see from Table 5 that the recognition result is not very high compared with the other target recognition results. There are several reasons for this. First, the detection datasets of this paper contained 30 different kinds of traffic signs. They are not simply detected based on the shapes of the traffic signs, but the specific semantic information is recognized. Second, the recognition scenes are traffic scenes in a real foggy environment. The foggy environment and the complex environments of the road bring great difficulty to the recognition of traffic signs. Finally, the focus of this paper was to improve the performance of the target detection algorithms and study the applicability of the defogging algorithms for traffic signs in real road scenes, aiming to prove that the combined algorithms proposed in this paper could effectively improve the recognition performance of YOLOv5 on the datasets of traffic signs in foggy environments. According to the data in the table, when detecting foggy traffic signs, the overall performance index of Pix2PixHD+YOLOv5-T is improved compared with the YOLOv5-T and YOLOv5 models. Among them, its precision reaches 78.5%, the recall reaches 72.2%, and the mAP@0.5 is 82.8%. mAP@0.5 is 3.8% higher than YOLOv5-T for detecting foggy images and 11.4% higher than YOLOv5 for detecting foggy images. The transformer module can fully improve the comprehensive detection performance of YOLOv5, and Pix2PixHD has a good defogging effect in the face of a real foggy environment. The experiments show that the YOLOv5-T model in this paper can make full use of advantages of the defogging algorithm Pix2PixHD, and the combination of the Pix2PixHD and YOLOv5-T models can effectively improve the recognition performance of traffic signs in a foggy environment.

## 6. Conclusions

Due to the inherent characteristics of traffic signs such as color, shape, and size, as well as the complex and changeable actual road environments, traffic sign recognition in foggy environments is very challenging. This paper proposed the Pix2PixHD+ YOLOv5-T algorithms by combining the defogging algorithm based on the Pix2PixHD network with the improved YOLOv5 algorithm to recognize traffic signs in foggy traffic scenes. First, the visualization tool TensorBoard was used to visualize the changing trends of the loss functions of the discriminator and generator in Pix2PixHD training, and the Pix2PixHD defogging method was compared with the mainstream defogging algorithms in qualitative and quantitative ways, which proved the applicability and effectiveness of the Pix2PixHD network in defogging traffic sign images in foggy traffic scenes. Then, the defogging algorithm based on the Pix2PixHD network was used to complete image defogging and achieve the conversion of foggy images to clear images. At the same time, the analysis revealed that the defogging images would be accompanied by the loss of object feature information and the reduction in pixel values, which provides a direction for the improvement of the detection algorithm. With this idea, the YOLOv5 recognition algorithm was improved to enhance YOLOv5′s ability to extract image feature information. Then, an improved model, YOLOv5-Transformer, was proposed to improve the efficiency and performance of the feature extraction of traffic signs with small targets. Based on the above study, Pix2PixHD was combined with YOLOv5-T to realize traffic sign detection in foggy traffic scenes. Finally, a visual comparative analysis of traffic sign recognition based on defogging algorithms was carried out on synthetic fog datasets and real fog datasets, and a comparative analysis of three recognition algorithms of traffic signs was carried out on moderate fog datasets. The comparative analysis results showed that whether in synthetic fog datasets or real fog datasets, compared with other methods, the Pix2PixHD+YOLOv5-T algorithm proposed in this paper has a good effect on traffic sign recognition in a foggy environment. However, this paper also has certain limitations. Only 30 types of traffic signs were selected for detection, and the types of detection could be further expanded. In addition, the optimization of the model was only local optimization; that is, the whole network could detect traffic signs in a foggy environment by improving the quality and speed of extracting target features from the local network. This feature extraction network still has numerous convolutional modules, so the loss of target features cannot be ignored. Therefore, the next stage of research and improvement goals include expanding the detection types and optimizing the feature extraction network to further improve the recognition performance of traffic signs in foggy environments.

## Figures and Tables

**Figure 1 sensors-24-04370-f001:**
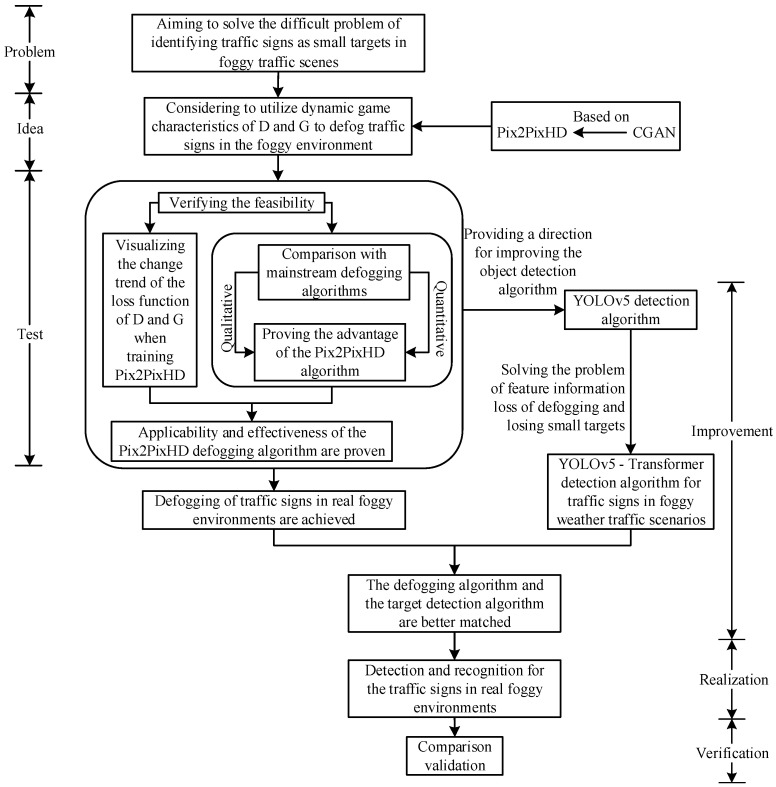
Overall research idea.

**Figure 2 sensors-24-04370-f002:**
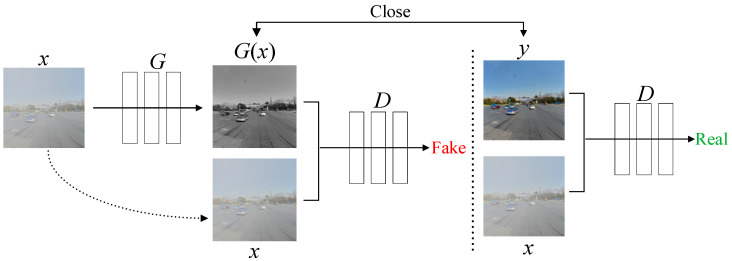
Conditional generative adversarial network structure.

**Figure 3 sensors-24-04370-f003:**
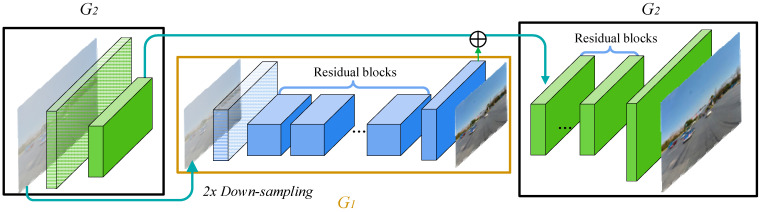
Architecture of the Pix2PixHD generator.

**Figure 4 sensors-24-04370-f004:**
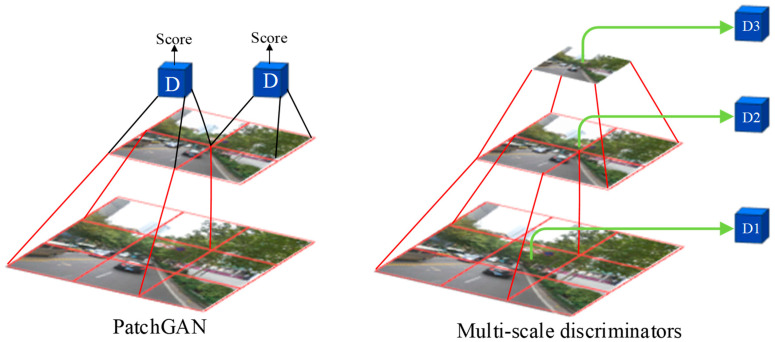
Architecture of the Pix2PixHD discriminator.

**Figure 5 sensors-24-04370-f005:**
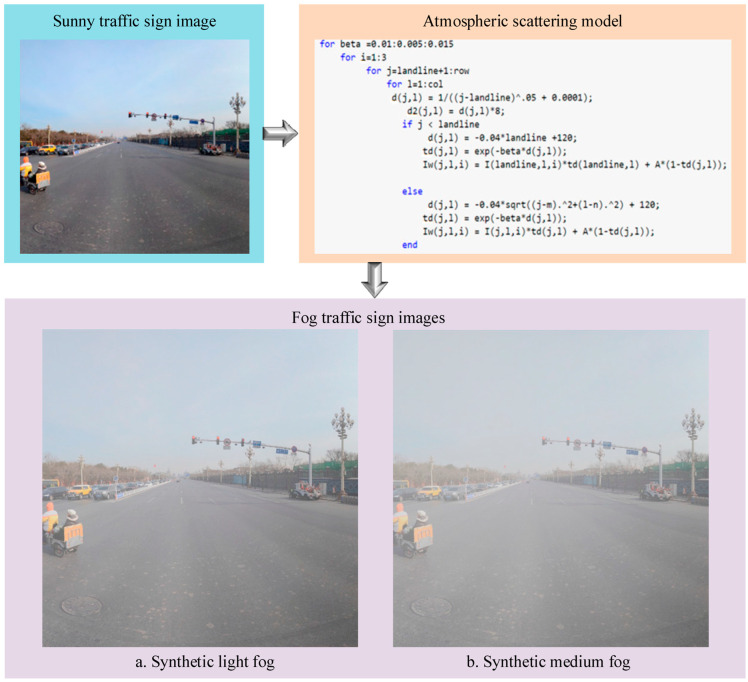
The construction process of a synthetic foggy image.

**Figure 6 sensors-24-04370-f006:**
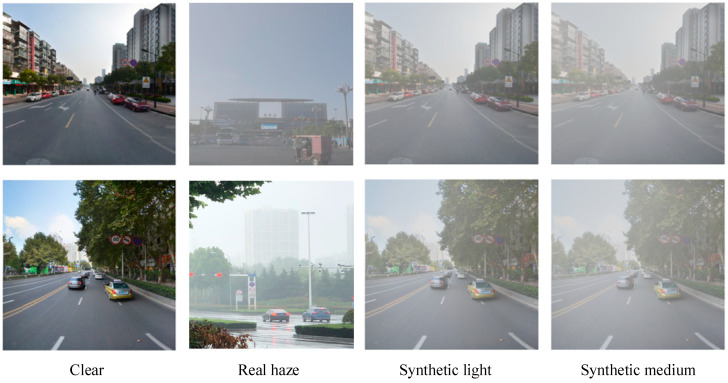
Image examples of the dataset constructed in this paper.

**Figure 7 sensors-24-04370-f007:**
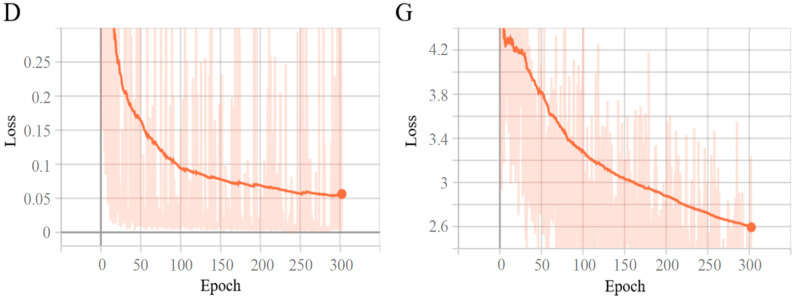
The trends of the loss function during training.

**Figure 8 sensors-24-04370-f008:**
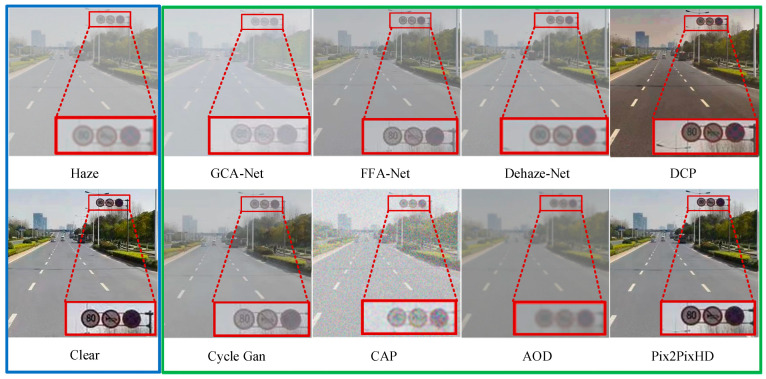
Qualitative analysis and visualization of defogging images.

**Figure 9 sensors-24-04370-f009:**
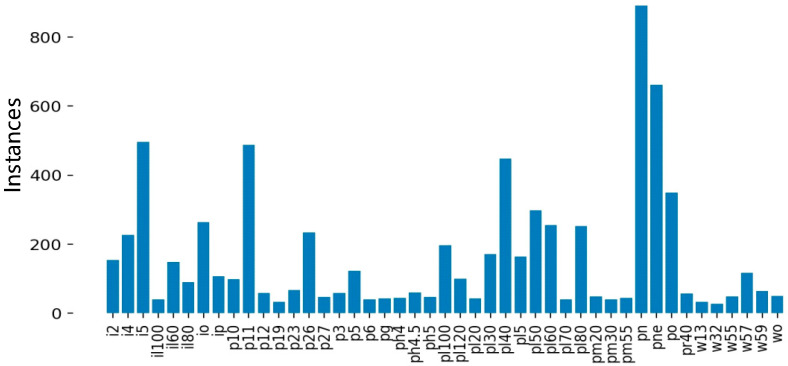
Types of traffic signs and data volume distribution before screening.

**Figure 10 sensors-24-04370-f010:**
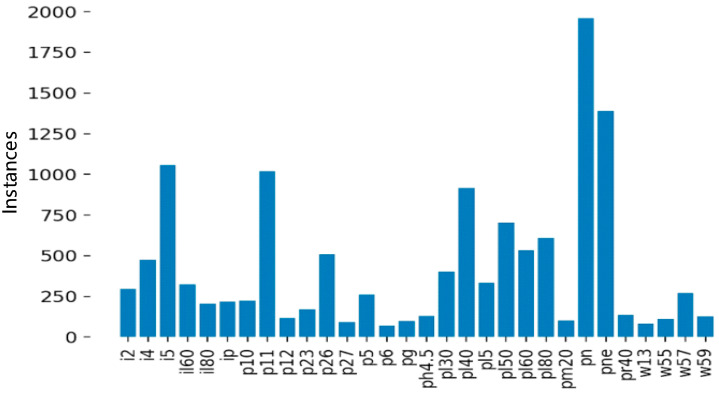
Types of traffic signs and data volume distribution after screening.

**Figure 11 sensors-24-04370-f011:**
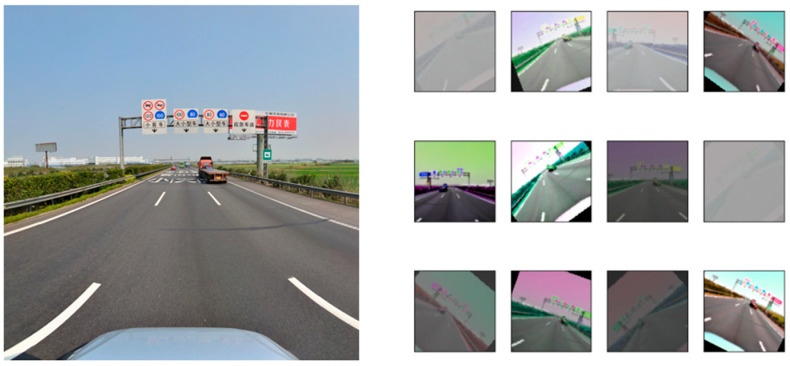
Sample of traffic sign image augmentation.

**Figure 12 sensors-24-04370-f012:**
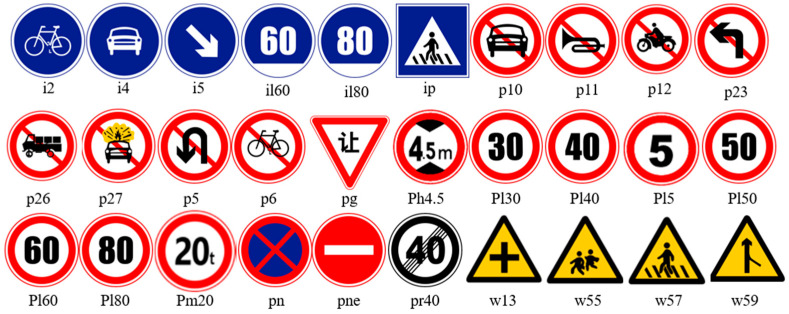
Thirty types of traffic signs.

**Figure 13 sensors-24-04370-f013:**
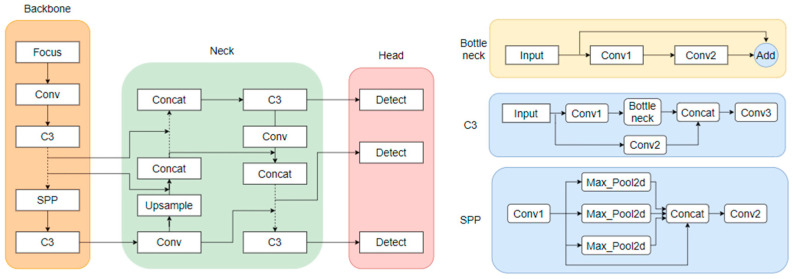
YOLOv5 model architecture and main module structure.

**Figure 14 sensors-24-04370-f014:**
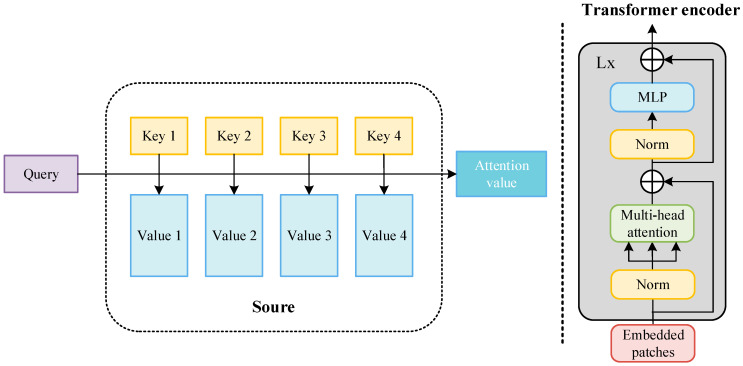
Components of the transformer’s architecture.

**Figure 15 sensors-24-04370-f015:**
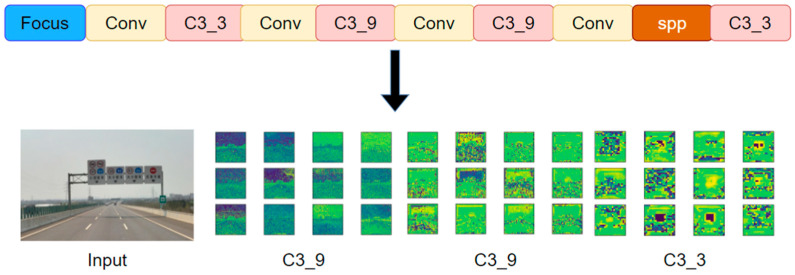
Feature extraction of traffic signs using the YOLOv5 backbone.

**Figure 16 sensors-24-04370-f016:**
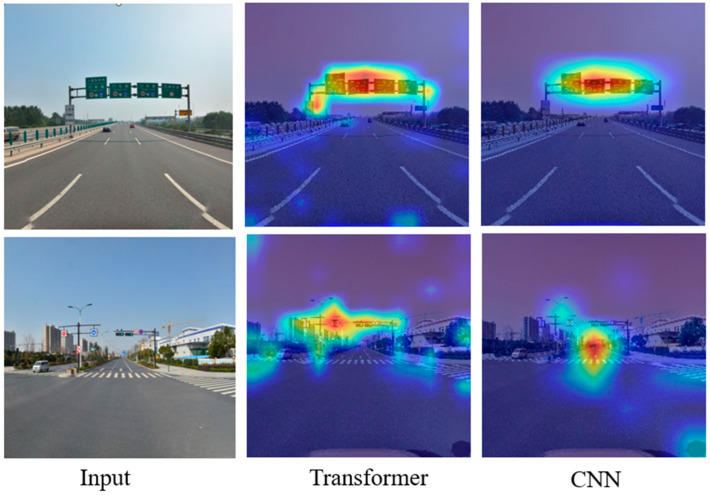
Thermograph comparison of a transformer and CNN.

**Figure 17 sensors-24-04370-f017:**
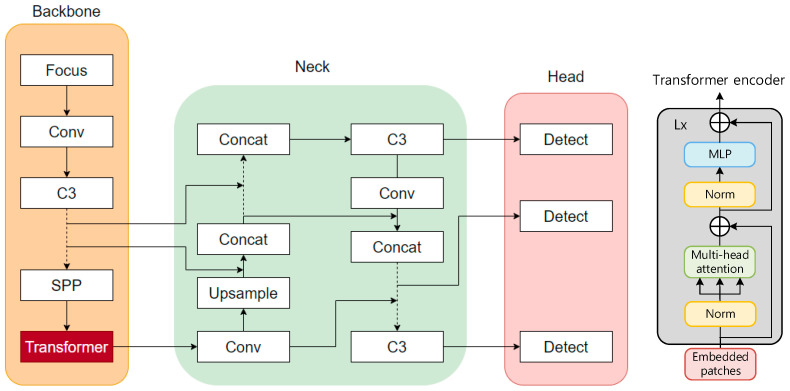
Architecture of YOLOv5 after the improvement.

**Figure 18 sensors-24-04370-f018:**
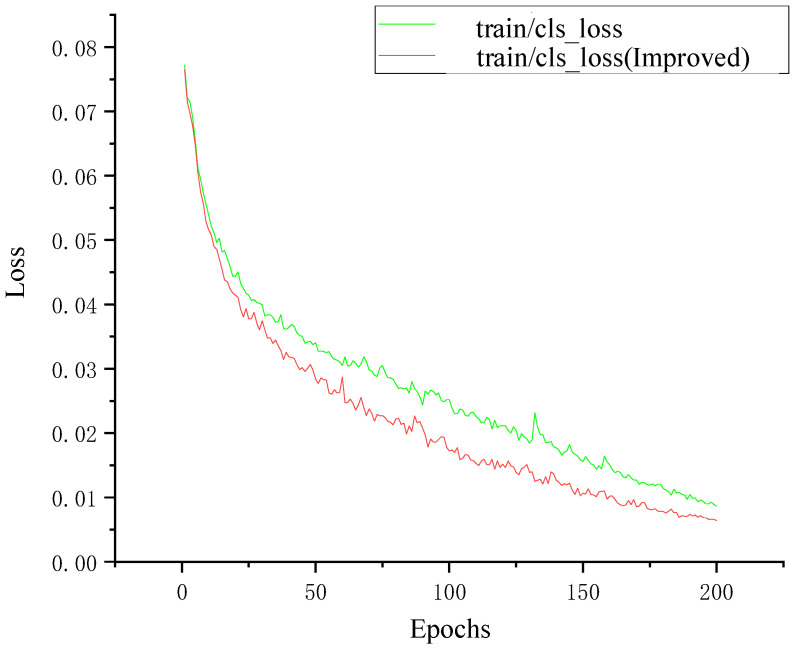
Training loss before and after the improvement.

**Figure 19 sensors-24-04370-f019:**
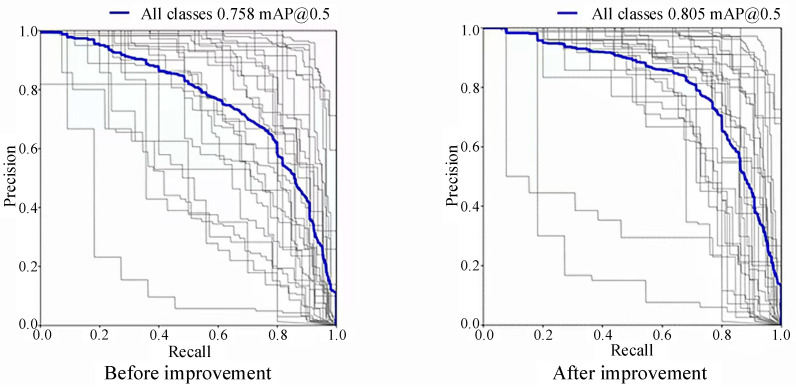
The comparison of mAP@0.5.

**Figure 20 sensors-24-04370-f020:**
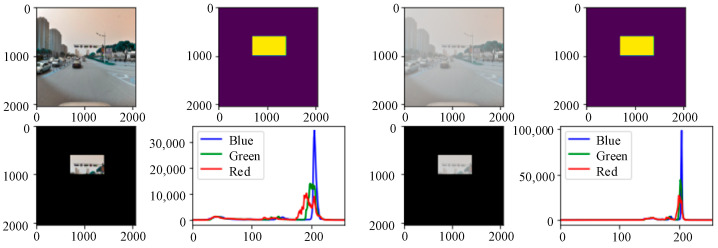
Color histograms of the detected targets in different environments.

**Figure 21 sensors-24-04370-f021:**
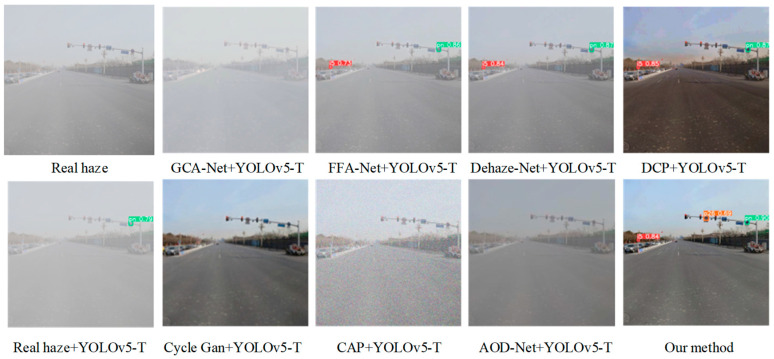
Recognition of traffic sign images in synthetic foggy environments.

**Figure 22 sensors-24-04370-f022:**
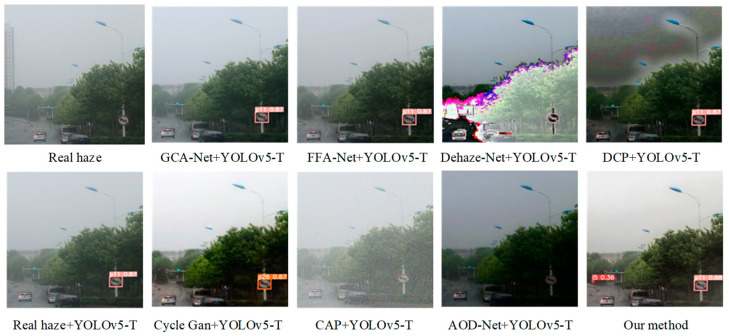
Recognition of traffic sign images in a real foggy environment.

**Table 1 sensors-24-04370-t001:** Training parameter settings.

Parameter	Value
Batch size	1
Input nc Output nc	3 3
Load size	1024
lr	0.0002
nite	150
niter_decay	150
save_epoch_freq	10

**Table 2 sensors-24-04370-t002:** Values of the SSIM and PSNR in light fog.

Light	SSIM	PSNR
GCA-Net	0.730	12.555
FFA-Net	0.858	20.617
Dehaze-Net	0.898	22.520
DCP	0.807	15.197
Cycle Gan	0.543	13.200
CAP	0.230	11.983
AOD-Net	0.573	13.800
Pix2PixHD	0.897	24.269

**Table 3 sensors-24-04370-t003:** Values of the SSIM and PSNR in medium fog.

Medium	SSIM	PSNR
GCA-Net	0.663	10.906
FFA-Net	0.931	22.603
Dehaze-Net	0.816	17.108
DCP	0.919	17.552
Cycle Gan	0.541	13.124
CAP	0.145	11.174
AOD-Net	0.594	14.093
Pix2PixHD	0.975	26.465

**Table 4 sensors-24-04370-t004:** Analysis of the model metrics at different visibility levels.

Measurement Indicator	Precision	Recall	*mAP*@0.5	*mAP*@0.5:0.95
Sunny	0.818	0.796	0.856	0.643
Medium fog	0.768	0.646	0.714	0.507

**Table 5 sensors-24-04370-t005:** Comparison of the recognition effects of three recognition methods for traffic signs.

Test Set	Algorithm	Precision	Recall	*mAP*@0.5	*mAP*@0.5:0.95
Sunny	YOLOv5	0.818	0.796	0.856	0.643
Medium fog	YOLOv5	0.768	0.646	0.714	0.507
Sunny	YOLOv5-T	0.787	0.815	0.852	0.646
Medium fog	YOLOv5-T	0.78	0.727	0.79	0.582
Medium fog	YOLOv5-T+Pix2PixHD	0.785	0.722	0.828	0.584

## Data Availability

The raw data supporting the conclusions of this article will be made available by the authors without undue reservation.

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
