# Peer review of "Research on a Recognition Algorithm for Traffic Signs in Foggy Environments Based on Image Defogging and Transformer"

_sensors, 2024, doi:10.3390/s24134370_

Round 1
Reviewer 1 Report
Comments and Suggestions for Authors
We appreciate the author's efforts in improving both the Image Defogging Algorithm and Traffic Sign Detection Algorithm. The manuscript meticulously presents comparative experiments and offers detailed analysis of the experimental outcomes.
However, it has come to my attention that there is complete duplication of paragraphs between sections L344-L378 and L610-L644. Additionally, there appears to be redundancy in Figures 11 and 23. These issues need to be addressed.
Author Response
Dear Reviewer,
Thank you for your kind comments concerning our manuscript (ID: sensors-3000872 Title: Research on Traffic Sign Detection Algorithm in Foggy Environment Based on Image Defogging and Transformer). We sincerely appreciate your valuable comments which are important to improve the quality of our manuscript. Your comments are showed in italicized font and specific concerns are numbered. Our point-to-point responses are given in blue text and all revisions for the manuscript are marked in yellow highlighting in the updated manuscript.
Comment: We appreciate the author's efforts in improving both the Image Defogging Algorithm and Traffic Sign Detection Algorithm. The manuscript meticulously presents comparative experiments and offers detailed analysis of the experimental outcomes.
However, it has come to my attention that there is complete duplication of paragraphs between sections L344-L378 and L610-L644. Additionally, there appears to be redundancy in Figures 11 and 23. These issues need to be addressed.
Response: We sincerely appreciate the valuable comment. Due to our mistake, we didn't present the manuscript in the best form. We have made the necessary modifications according to your suggestions. Furthermore, we proofread the manuscript carefully and made ensure that all issues had been resolved. Thanks again for your valuable comments.
Thank you for your kind comments.
Sincerely,
Authors
Corresponding author:
Name: Zhaohui Liu
E-mail: zhaohuiliu@sdust.edu.cn

Reviewer 2 Report
Comments and Suggestions for Authors
Dear colleagues, the research topic is of a great importance due to a challenging task of traffic signs recognition. Although this problem has been addressed in many papers, still there is no a "perfect" solution. The proposed approach based on a generative pix2pixHD framework to build a defogging model paves a new way to achieve human-like quality of signs recognition.
I would recommend the manuscript to be accepted with no remarks.
Author Response
Dear Reviewer,
Thank you for your kind comments concerning our manuscript (ID: sensors-3000872 Title: Research on Traffic Sign Detection Algorithm in Foggy Environment Based on Image Defogging and Transformer). We sincerely appreciate your valuable comments which are important to improve the quality of our manuscript. Your comments are showed in italicized font and specific concerns are numbered. Our point-to-point responses are given in blue text and all revisions for the manuscript are marked in yellow highlighting in the updated manuscript.
Comment: Dear colleagues, the research topic is of a great importance due to a challenging task of traffic signs recognition. Although this problem has been addressed in many papers, still there is no a "perfect" solution. The proposed approach based on a generative Pix2PixHD framework to build a defogging model paves a new way to achieve human-like quality of signs recognition.
I would recommend the manuscript to be accepted with no remarks.
Response: We deeply appreciate your kind comments. Thank you very much for your approval of our research. In the future, we will continue to work hard to overcome the puzzle of detection of traffic signs in adverse weather.
Thank you for your kind comments.
Sincerely,
Authors
Corresponding author:
Name: Zhaohui Liu
E-mail: zhaohuiliu@sdust.edu.cn
Reviewer 3 Report
Comments and Suggestions for Authors
This manuscript presents a traffic sign detection algorithm for foggy environment based on image defogging and transformer. Although the research results seem significant, the logic structure of the manuscript is quite confusing.
(1) The method proposed in the paper is not presented as a whole. From the three section titles, " 2. Image defogging", " 3. Training experiments based on pix2pixHD network ", and " 4. Traffic Sign Recognition", readers cannot get the proposed overall method.
(2) The authors do not highlight their improved modules. For example, in abstract section, the authors declare that "Inspired by the literature [28], this paper adapts the data reading module of the Pix2PixHD network and makes foggy and sunny day images to train the network, to realize the conversion of foggy day images to sunny day images. ", but Pix2PixHD with conditional GANs is a published well method [Ting-Chun Wang, Ming-Yu Liu, Jun-Yan Zhu, Andrew Tao, Jan Kautz, Bryan Catanzaro. High-Resolution Image Synthesis and Semantic Manipulation with Conditional GANs, CVPR 2018.]. The authors are not inspired by [28], but the Pix2PixHD algorithm should be your baseline.
(3) On image defogging, the experimental results that should be compared are typical Pix2PixHD and your improved Pix2PixHD algorithms.
(4) The introduction of experimental data is vague and lacks a description of the amount of experimental data, image size, and experimental process.
(5) The manuscript's title uses traffic sign detection, but the fourth section focuses on traffic signal recognition. Undoubtedly, there are significant differences in the purpose and evaluation mechanism of detection and recognition.
(6) The experimental parameters were given without explanation. For example, on page 4, "the atmospheric light value A was set to 0.8 and the scattering rate β to 0.04 and 0.08, respectively, using MATLAB based on the principle of atmospheric scattering model to simulate 160 the light fog and medium fog. ". How to add fog should also be clearly described or cited.
(7) Many descriptions are insufficient evidence or may cause controversy. For example, "With the increase of pollution caused by industrial production and transportation, foggy weather occurs frequently in cities[1]. ". This statement is neither the original sentence nor can be inferred from this article [1].
Comments on the Quality of English LanguageMust be improved.
Author Response
Dear Reviewer,
Thank you for your kind comments concerning our manuscript (ID: sensors-3000872 Title: Research on Traffic Sign Detection Algorithm in Foggy Environment Based on Image Defogging and Transformer). We sincerely appreciate your valuable comments which are important to improve the quality of our manuscript. Your comments are showed in italicized font and specific concerns are numbered. Our point-to-point responses are given in blue text and all revisions for the manuscript are marked in yellow highlighting in the updated manuscript.
Comment 1: The method proposed in the paper is not presented as a whole. From the three section titles, "2. Image defogging", "3. Training experiments based on Pix2PixHD network", and "4. Traffic Sign Recognition", readers cannot get the proposed overall method.
Response: We think this is an excellent suggestion, which will greatly help to improve the overall level of our manuscript. We have made improvements from two aspects: On the one hand, the research idea diagram is added in the Part 1 Introduction, which makes readers can clearly understand the overall research idea of this manuscript; On the other hand, we have revised the titles of Parts 2 to 5 and adjusted the content structure. The title is modified as follows: 2. Principle of Defogging Algorithm Based on Conditional Generative Adversarial Network, this part mainly introduces the principle of the improved - Pix2PixHD network based on Conditional Generative Adversarial Network (CGAN) and its application in defogging of traffic sign image in fog environment; 3. The Training Experiment Based on Pix2PixHD Network, this part mainly introduces that the images in traffic sign Dataset of sunny day are fogged according to the atmospheric scattering model, and then the Dataset of paired sunny day image and foggy day image is generated, and the Pix2PixHD defogging algorithm is trained. The change trend of loss function of Pix2PixHD network is visualized. It is compared qualitatively and quantitatively with a variety of mainstream defogging algorithms to prove the applicability and effectiveness of applying Pix2PixHD network to defogging traffic sign images in traffic scenes of foggy day. Then, the defogging algorithm based on Pix2PixHD network is adopted for defogging of traffic sign images in fog environment; 4. The Detection Algorithm for Traffic Sign Based on Improved YOLOv5, this part mainly introduces that the improvement orientation of YOLOv5 is determined based on the training results of Pix2PixHD network and the results of comparison with a variety of mainstream defogging algorithms, as a result, the YOLOv5-Transformer (namely YOLOv5-T) detection algorithm is determined; 5. Recognition of Traffic Sign in Foggy Environment and Contrastive Analysis of Recognition Effect, this part mainly introduces that the image defogging algorithm based on Pix2PixHD network and the YOLOv5-T detection algorithm are combined to identify traffic signs in traffic scenes of foggy day, as well as the comparative analysis of different methods are carried out which verifies the performance advantages of Pix2PixHD+YOLOv5-T algorithm in detection of traffic sign in fog environment. Specific modifications are highlighted in the manuscript.
Comment 2: The authors do not highlight their improved modules. For example, in abstract section, the authors declare that "Inspired by the literature [28], this paper adapts the data reading module of the Pix2PixHD network and makes foggy and sunny day images to train the network, to realize the conversion of foggy day images to sunny day images. ", but Pix2PixHD with conditional GANs is a published well method [Ting-Chun Wang, Ming-Yu Liu, Jun-Yan Zhu, Andrew Tao, Jan Kautz, Bryan Catanzaro. High-Resolution Image Synthesis and Semantic Manipulation with Conditional GANs, CVPR 2018.]. The authors are not inspired by [28], but the Pix2PixHD algorithm should be your baseline.
Response: Thank you for your question. Sorry, we didn't make it clear. Enlightens by literature [28], Pix2PixHD network is applied to defog the traffic sign image in fog environment by utilizing the characteristics of dynamic game of discriminator and generator in conditional generation adversarial network. The defogging algorithm of Pix2PixHD is trained, and the change trend of loss function of Pix2PixHD network is visualized. The qualitative and quantitative comparison is made with the mainstream defogging algorithm, which proves the applicability and effectiveness of applying Pix2PixHD network to defogging the images of traffic signs in foggy traffic scenes, as well as provides an evidence for selection of image defogging algorithm of traffic sign in foggy traffic scenes.
Comment 3: On image defogging, the experimental results that should be compared are typical Pix2PixHD and your improved Pix2PixHD algorithms.
Response: Thank you for your suggestion. This manuscript focuses on the improvement of YOLOv5 model and the combination of Pix2PixHD with YOLOV5-T to realize the accurate detection and recognition of traffic signs in foggy traffic scenes. This manuscript doesn’t improve Pix2PixHD network, but utilize the characteristics of dynamic game of discriminator and generator in conditional generation adversarial network to defogging the images of traffic signs in foggy traffic scenes. In order to verify its feasibility, the change trend of loss function of Pix2PixHD network is visualized, and the qualitative and quantitative comparisons are made with the mainstream defogging algorithm. Based on the training of Pix2PixHD network and the comparative analysis with mainstream defogging algorithms, the improvement orientation of YOLOv5 target detection algorithm is determined, so that the defogging algorithm based on Pix2PixHD network can be better matched with the improved YOLOv5 model, and the detection and recognition of traffic signs, a small target, can be improved in foggy traffic scenes. The optimization of Pix2PixHD proposed by you will be taken as the research direction in the future, and we will strive to make further contributions to improve the accuracy and efficiency of detection for traffic signs in adverse weather. Thank you again.
Comment 4: The introduction of experimental data is vague and lacks a description of the amount of experimental data, image size, and experimental process.
Response: We sincerely appreciate the valuable comment. The constructive process of synthetic fog image has been added in Section 3.1 Building Training Dataset; In the Section 2.2.1 Pix2PixHD Generator Structure, input and output image sizes of G1 and G2 are 1024×512 and 2048×1024, respectively; The analysis of RGB three-channel histogram is added to Section 5.1 Influence of Fog on the Image Recognition of Traffic Sign, and the difference of color saturation in the RGB histogram of sunny and foggy images is explained. Specific modifications are highlighted in the manuscript.
Comment 5: The manuscript's title uses traffic sign detection, but the fourth section focuses on traffic signal recognition. Undoubtedly, there are significant differences in the purpose and evaluation mechanism of detection and recognition.
Response: Thank you for your valuable comments. We have modified the original title 4.Traffic Sign Recognition to the title 4.The Detection Algorithm for Traffic Sign Based on Improved YOLOv5.
Comment 6: The experimental parameters were given without explanation. For example, on page 4, "the atmospheric light value A was set to 0.8 and the scattering rate β to 0.04 and 0.08, respectively, using MATLAB based on the principle of atmospheric scattering model to simulate the light fog and medium fog.". How to add fog should also be clearly described or cited.
Response: Thank you for your suggestion. ①The parameters are determined according to the previous research of our research group, please refer to the reference
Runze Song, Zhaohui Liu*, Chao Wang. End-to-end dehazing of traffic sign images using reformulated atmospheric scattering model[J]. Journal of Intelligent & Fuzzy Systems, 2021, 41 (6): p6815-6830.
We have added this reference to our manuscript.
②The constructive process of synthetic fog image has been added in Section 3.1 Building Training Dataset. Specific modifications are highlighted in the manuscript.
Comment 7: Many descriptions are insufficient evidence or may cause controversy. For example, "With the increase of pollution caused by industrial production and transportation, foggy weather occurs frequently in cities[1]. ". This statement is neither the original sentence nor can be inferred from this article [1].
Response: Thank you for your valuable comments, which helps use to improve the overall level of our manuscript. We have added explanations for unclear content and fixed errors. Reference [1] has been revised to quote the original sentence of its Abstract.
Thank you for your kind comments.
Sincerely,
Authors
Corresponding author:
Name: Zhaohui Liu
E-mail: zhaohuiliu@sdust.edu.cn
Reviewer 4 Report
Comments and Suggestions for Authors
This paper modifies the conditional generative adversarial network (pix2pixHD) to create a defogging model suitable for traffic sign recognition tasks and introduces the Transformer module into Yolov5 to enhance its feature extraction capabilities. The improved defogging and detection algorithms are then combined for traffic sign recognition in foggy environments. Although the article is rich in content, it does not meet the standards of this journal for the following reasons:
- The literature review in Section 1 is insufficient. There have been many studies related to traffic sign recognition in adverse weather conditions such as fog, but the literature mentioned in the introduction is not comprehensive.
- There are numerous misuses of abbreviations in the article. Abbreviations need only be given their full names when they first appear in the text and can be quoted directly thereafter. For example, the definition of PSNR is provided on line 250 of the paper, and only PSNR should be used on lines 274 and 672. Similar issues exist with abbreviations such as FPN and SSIM.
- Authors should carefully read reference 15 to confirm whether "Yolov3" in line 72 is correct.
- "PSRN" in the titles and content of Tables 2 and 3 should be "PSNR."
- "PSRN" in lines 289, 290, 297, and 302 should also be "PSNR."
- The description of the dataset used in this paper is vague. In section 4.1, please add references to the datasets GTSDB, BDD100K, Oxford, RobotCar Dataset, and ApolloScape. Specify how and how many of these datasets were used in this article.
- Figures 8 and 7 are identical. What are the screening rules?
- Line 338 claims to have selected 30 different types of traffic signs. How many samples are there for each category?
- Line 366 states that there are 7000 traffic sign images under clear sky conditions. Explain the number of images for each category. Additionally, provide a more detailed introduction of the foggy dataset.
- The definition of FPN in lines 352 and 353 is inconsistent.
- There are many formatting errors in the manuscript. For example, line 353 has an extra ")". Careful inspection is required.
- There are many grammatical errors in the manuscript. For example, there are syntax errors on lines 372, 373, and 639. Careful inspection is required.
- Remove an extra "of the" from line 464.
- Confirm whether the unit of the X-axis Epoch in Figure 5 is "K".
- Confirm the label of the Y-axis in Figure 16.
- The data for the mAP metric in Table 5 is clearly not a percentage.
- The description of the results data in lines 640-641 is vague and cannot be understood.
- "map" in line 658 should be "mAP."
- Figures 21 and 11 are exactly the same.
- The author only compared the defogging algorithm proposed in this paper with existing defogging algorithms but did not compare Yolov5-T+Pix2pixHD with other existing traffic sign detection algorithms. Please add a comparative experiment.
Author Response
Dear Reviewer,
Thank you for your kind comments concerning our manuscript (ID: sensors-3000872 Title: Research on Traffic Sign Detection Algorithm in Foggy Environment Based on Image Defogging and Transformer). We sincerely appreciate your valuable comments which are important to improve the quality of our manuscript. Your comments are showed in italicized font and specific concerns are numbered. Our point-to-point responses are given in blue text and all revisions for the manuscript are marked in yellow highlighting in the updated manuscript.
Comment 1: The literature review in Section 1 is insufficient. There have been many studies related to traffic sign recognition in adverse weather conditions such as fog, but the literature mentioned in the introduction is not comprehensive.
Response: We appreciate the reviewer’s insightful and constructive comment.We added literature 24,25,26 in the "1. Introduction" to compensate for the deficiencies in the literature. See line 99-106 for specific changes.
Comment 2: There are numerous misuses of abbreviations in the article. Abbreviations need only be given their full names when they first appear in the text and can be quoted directly thereafter. For example, the definition of PSNR is provided on line 250 of the paper, and only PSNR should be used on lines 274 and 672. Similar issues exist with abbreviations such as FPN and SSIM.
Response: Thank you for your kind comment. We have modified them accordingly.
Comment 3: Authors should carefully read reference 15 to confirm whether "Yolov3" in line 72 is correct.
Response: Thank you for your positive feedback. Sorry due to the input error as we edited the manuscript. ''Yolov5'' is described in reference 15. Thank you for correcting our manuscript.
Comment 4: "PSRN" in the titles and content of Tables 2 and 3 should be "PSNR."
Response: Thank you for pointing out the error in Table 2 and Table 3 where "PSRN" should be "PSNR". We have corrected it, thanks for your valuable feedback.
Comment 5: "PSRN" in lines 289, 290, 297, and 302 should also be "PSNR".
Response: Thank you for your valuable feedback. We have modified them accordingly.
Comment 6: The description of the dataset used in this paper is vague. In section 4.1, please add references to the datasets GTSDB, BDD100K, Oxford RobotCar Dataset, and ApolloScape. Specify how and how many of these datasets were used in this article.
Response: Thank you for your valuable comments. In Section 4.1 of this article, We have supplemented the literature 38, 39, 40, 41 on the GTSDB, BDD100K, Oxford RobotCar Dataset, and ApolloScape. We picked the pictures suitable for the experiment to construct the dataset. About 6000 copies were picked from the TT100K Dataset and the GTSDB Dataset. BDD100K, Oxford RobotCar Dataset, ApolloScape, etc., a total of 1000 Haze images were selected in these alternative Datasets.
Comment 7: Figures 8 and 7 are identical. What are the screening rules?
Response: Thank you for your kind comment. Due to article modification issues, the original Figures 7 and 8 correspond to Figures 9 and 10 of the present article. Figure 9 shows 45 types of traffic signs collected after data expansion and screening. However, some traffic signs (e. g., il100, p19, w32) appear less than 100 times in the Dataset, meaning that they are not frequent signs. This results in machine learning models being unable to adequately learn the features of these traffic signs.Therefore, the collected images were further enlarged to remove the underperforming samples from the dataset. Finally, 30 kinds of traffic signs were selected, namely Figure 10.
Comment 8: Line 338 claims to have selected 30 different types of traffic signs. How many samples are there for each category?
Response: Thank you for your kind comment. Each image contains multiple traffic signs. It is impossible to determine the exact number of pictures of each type of traffic sign. In total, we used 7,000 images to construct the dataset.
Comment 9: Line 366 states that there are 7000 traffic sign images under clear sky conditions. Explain the number of images for each category. Additionally, provide a more detailed introduction of the foggy Dataset.
Response: Thanks again for your careful review. Because each image may contain multiple traffic signs, it is impossible to accurately calculate the number of images for each category. It can only be estimated from a total number of 7,000 images. When selecting data, we focuses on picture quality rather than quantity. Our goal is to build a new Dataset that can meet the needs. Although specific quantities are not particularly emphasized, the focus is on improving the overall quality and applicability of the dataset.
Comment 10: The definition of FPN in lines 352 and 353 is inconsistent.
Response: Thank you for your kind comment. We have changed it accordingly.
Comment 11: There are many formatting errors in the manuscript. For example, line 353 has an extra ")". Careful inspection is required.
Response: Thank you for your careful inspection of our manuscript. We have removed ")".
Comment 12: There are many grammatical errors in the manuscript. For example, there are syntax errors on lines 372, 373, and 639. Careful inspection is required.
Response: Thank you for your constructive criticism. We are very sorry for the trouble caused by our presentation. We have revised it back here.
Comment 13: Remove an extra "of the" from line 464.
Response: Thank you for your careful inspection of our manuscript. We have removed "of the".
Comment 14: Confirm whether the unit of the X-axis Epoch in Figure 5 is "K".
Response: Thanks for your careful review of the manuscript. Due to article modification issues, the original Figures 5 correspond to Figures 7 of the present article. Regarding the unit question of the X-axis Epoch in Figure 7, it is confirmed that the unit of Epoch is not 'K', but directly counted by the number of epochs. We have revised Figure 7.
Comment 15: Confirm the label of the Y-axis in Figure 16.
Response: Thank you very much for your valuable comment. Due to article modification issues, the original Figures 16 correspond to Figures 18 of the present article. We have carefully examined and confirmed the Y-axis label in Figure 16. We changed the original Y-axis label "train / cls _ loss" to "Loss".
Comment 16: The data for the mAP metric in Table 5 is clearly not a percentage.
Response: Thank you for your kind comment. We have now cut out the extra "/%".
Comment 17: The description of the results data in lines 640-641 is vague and cannot be understood.
Response: Thank you for reviewing the article. We are sorry for our lack of English expression. We have reformulated this section.
Comment 18: "map" in line 658 should be "mAP."
Response: Thank you for your kind comment. We have modified the "map" to the "mAP".
Comment 19: Figures 21 and 11 are exactly the same.
Response: Thank you for the detailed inspection. We have removed the excess of Figure 21.
Comment 20: The author only compared the defogging algorithm proposed in this paper with existing defogging algorithms but did not compare Yolov5-T+Pix2PixHD with other existing traffic sign detection algorithms. Please add a comparative experiment.
Response: Thank you for your question. Maybe because we did not make it clear, you did not notice our comparison experiment. We have revised the expression to highlight the comparative analysis of different methods from different angles. In actual traffic scenarios, traffic signs are small targets compared with other detection targets. Therefore, the missing detection,wrong detection and low accuracy are common problems at present. For the above problems, the main task of this article is to improve the accuracy and efficiency of traffic sign identification in foggy environment by combining the defogging algorithm based on Pix2PixHD network and the improved Yolov5 algorithm. It reflects the comprehensive advantages of the combination of the two to identify the traffic sign in the actual traffic scene. In order to verify the superiority of the proposed algorithm Pix2PixHD + Yolov5-T, the recognition effect of the synthetic traffic sign image in the actual traffic scene and the real traffic sign image based on the fog removal algorithm. On this basis, in order to further prove the superiority of the algorithm, the images of sunny day traffic sign and real foggy traffic image are used as the test set to further identify the Yolov5, Yolov5-T and Pix2PixHD + Yolov5-T algorithms involved in this paper. The validity and applicability of the algorithm Pix2PixHD + Yolov5-T for traffic sign detection in fog days are confirmed.
Thank you for your kind comments.
Sincerely,
Authors
Corresponding author:
Name: Zhaohui Liu
E-mail: zhaohuiliu@sdust.edu.cn
Round 2
Reviewer 3 Report
Comments and Suggestions for Authors
The author has made some modifications according to the review comments, but the submitted manuscript still needs further revisions.
1. The sentence "which are 1.7%, 7.6%, and 11.4%, respectively, higher than those of the algorithm that directly detects foggy images" in Section Abstract, is ambiguous.
2. Regarding Figure 1, I suggested last time that the process of the proposed algorithm be provided rather than the structure of the manuscript.
3. Regarding experimental data, it is necessary to provide the number of training and testing images.
4. In Figure 3, the formula "2x", is incorrect, because "x" is an image in Figure 2, but "2x" is not defined. Please refer to the reference on The Pix2PixHD model.
5. Please provide enlarged views of the local area marked with red boxes in Figure 8.
6. It is hoped that the test results of the proposed algorithm, Yolov5-T+Pix2PixHD, on this dataset, Sunny, can be added to Table 5.
7. Overall, this paper does not describe the proposed algorithm and experiments in detail, making it difficult for readers to reproduce them accurately.
Comments on the Quality of English Language
Moderate editing of English language required.
Author Response
Dear Reviewer,
Thank you for your kind comments concerning our manuscript (ID: sensors-3000872 Title: Research on Traffic Sign Detection Algorithm in Foggy Environment Based on Image Defogging and Transformer). We sincerely appreciate your valuable comments which are important to improve the quality of our manuscript. Your comments are showed in italicized font and specific concerns are numbered. Our point-to-point responses are given in blue text and all revisions for the manuscript are marked in yellow highlighting in the updated manuscript.
Comment 1: The sentence "which are 1.7%, 7.6%, and 11.4%, respectively, higher than those of the algorithm that directly detects foggy images" in Section Abstract, is ambiguous.
Response: Thank you for your valuable suggestion. We apologize for any confusion caused by our unclear expression. We have now revised the expression approach.
Comment 2: Regarding Figure 1, I suggested last time that the process of the proposed algorithm be provided rather than the structure of the manuscript.
Response: We sincerely appreciate the valuable comment and your kind reminder. We have revised Figure 1 accordingly. More clear illustration about the overall research ideation of the manuscript and the process of the proposed algorithm be provided.
Comment 3: Regarding experimental data, it is necessary to provide the number of training and testing images.
Response: Thank you for your valuable suggestions. On the basis of the dataset of 7000 images, we have divided the dataset into a training set, a validation set, and a test set according to the ratio of 8:1:1. We have clarified this in line 403-404 of the manuscript.
Comment 4: In Figure 3, the formula "2x", is incorrect, because "x" is an image in Figure 2, but "2x" is not defined. Please refer to the reference on The Pix2PixHD model.
Response: We think this is an excellent suggestion, and sincerely thank you for your careful review. We have redefined "2x". The input fog image is double downsampling via convolution layer of a generator G2, and then another generator G1 is used to generate low-resolution images. The output generated by G1 and the image obtained from the down-sampling are carried out element-wise adding, and then this combined result is fed into the subsequent network of G2 to generate high-resolution images. In the generator structure of Pix2PixHD, 2x down-sampling refers to reducing the width and height of the input image to half of its original size through a convolutional layer with a step size of 2, in order to extract the features of images on different scales and reduce the computational load for subsequent processing.
Comment 5: Please provide enlarged views of the local area marked with red boxes in Figure 8.
Response: Thank you for your friendly comment. We have revised Figure 8 as requested.
Comment 6: It is hoped that the test results of the proposed algorithm, Yolov5-T+Pix2PixHD, on this dataset, Sunny, can be added to Table 5.
Response: Thank you for your kind comment. Since the Pix2PixHD+Yolov5-T algorithm was originally designed to solve the problem of traffic sign recognition in real foggy weather images, the Sunny dataset, which represents clear-weather images with already high-quality clarity, doesn't require defogging processing. In practical applications, applying a defogging algorithm like Pix2PixHD to clear images may not significantly improve performance but could potentially reduce detection speed due to additional computational overhead. Therefore, we believe that conducting an experiment about the Yolov5-T+Pix2PixHD algorithm on the Sunny dataset is neither practical nor necessary. This manuscript focuses on the algorithm matching and performance of the Pix2PixHD+Yolov5-T algorithm for recognition of traffic signs of complex traffic scenario in foggy environment, and we have conducted experiments and verifications on relevant foggy weather datasets. These experimental results are sufficient to demonstrate the advantages and effects of the Pix2PixHD+Yolov5-T algorithm under foggy environment.
Comment 7: Overall, this paper does not describe the proposed algorithm and experiments in detail, making it difficult for readers to reproduce them accurately.
Response: We appreciate the reviewer's insightful and constructive comments, and we have carefully addressed these concerns and made a proper revision of the manuscript, which have been greatly helpful in improving the overall quality of our paper. Based on your feedback, we have revised the Abstract and redrawn the algorithm flowchart to more clearly illustrate the overall research ideation of the manuscript and the process of the proposed algorithm be provided. Through two major revisions of the manuscript, additional explanatory notes for relevant content, and repeated refinement of the English text, we have tried our best to describe the algorithm and experiments in this paper. We hope it meets your satisfaction.
Thank you for your kind comments.
Sincerely,
Authors
Corresponding author:
Name: Zhaohui Liu
E-mail: zhaohuiliu@sdust.edu.cn
Reviewer 4 Report
Comments and Suggestions for Authors
Figure 10 is still the same as Figure 9. It still contains 45 traffic signs categories, not 30 claimed by the authors.
Author Response
Dear Reviewer,
Thank you for your kind comments concerning our manuscript (ID: sensors-3000872 Title: Research on Traffic Sign Detection Algorithm in Foggy Environment Based on Image Defogging and Transformer). We sincerely appreciate your valuable comments which are important to improve the quality of our manuscript. Your comments are showed in italicized font and specific concerns are numbered. Our point-to-point responses are given in blue text and all revisions for the manuscript are marked in yellow highlighting in the updated manuscript.
Comment: Figure 10 is still the same as Figure 9. It still contains 45 traffic signs categories, not 30 claimed by the authors.
Response: We were really sorry for our careless mistakes. Thank you for your careful review. We have carefully checked the submitted manuscript again and realized that we failed to save the replaced Figure 10. We have now made the necessary revisions. Once again, we apologize for our oversight.
Thank you for your kind comments.
Sincerely,
Authors
Corresponding author:
Name: Zhaohui Liu
E-mail: zhaohuiliu@sdust.edu.cn
Round 3
Reviewer 3 Report
Comments and Suggestions for Authors
This paper has been significantly modified according to the review comments, making it a relatively complete work. However, there are still a few minor issues:
- On page 13, the word "Enlargement" in "Figure 11. Sample Traffic Sign Enlargement." is inexact.
- The full-text description and English expression need further careful revision. For example, "this paper applies Pix2PixHD network to remove the fog of traffic sign images in fog traffic scenes" in lines 129-130; "and d、d k、dv is the vector dimension" in line 450...
- Is this the Principle or Structure in Figure 2. Principle of Conditional Generative Adversarial Network Structure ?
Extensive editing of English language required.
Author Response
Dear Reviewer,
Thank you for your kind comments concerning our manuscript (ID: sensors-3000872 Title: Research on Traffic Sign Detection Algorithm in Foggy Environment Based on Image Defogging and Transformer). We sincerely appreciate all of reviewers for your valuable comments which are important to improve the quality of our manuscript. The reviewer’ comments are showed in italicized font and specific concerns are numbered. Our point-to-point responses are given in blue text and all revisions for the manuscript are marked in yellow highlighting in the updated manuscript.
Response to Reviewer 3
Comment 1: On page 13, the word "Enlargement" in "Figure 11. Sample Traffic Sign Enlargement." is inexact.
Response: We sincerely appreciate your hard work and kind reminder. We have revised the relevant expression.
Comment 2: The full-text description and English expression need further careful revision. For example, "this paper applies Pix2PixHD network to remove the fog of traffic sign images in fog traffic scenes" in lines 129-130; "and d、d k、dv is the vector dimension" in line 450...
Response: Thank you for your valuable suggestion. We apologize for any confusion caused by our unclear expression. We have revised the English expression accordingly and double checked the overall content of the manuscript. We have tried our best to polish the language, as well as have modified English words and grammar. The modifications help the content of the manuscript to be better understood. And here we didn’t list the modification, but marked them with yellow highlight in the revised manuscript.
Comment 3: Is this the Principle or Structure in Figure 2. Principle of Conditional Generative Adversarial Network Structure ?
Response: Thank you for your thoughtful suggestion. Now we have modified it to "Conditional Generation Adversarial Network structure".
Thank you again and best regards!
Sincerely,
Authors
Corresponding author:
Name: Zhaohui Liu
E-mail: zhaohuiliu@sdust.edu.cn